# Robust Autism Spectrum Disorder Screening Based on Facial Images (For Disability Diagnosis): A Domain-Adaptive Deep Ensemble Approach

**DOI:** 10.3390/diagnostics15131601

**Published:** 2025-06-24

**Authors:** Mohammad Shafiul Alam, Muhammad Mahbubur Rashid, Ahmad Jazlan, Md Eshrat E. Alahi, Mohamed Kchaou, Khalid Ayed B. Alharthi

**Affiliations:** 1Department of Mechatronics Engineering, International Islamic University Malaysia, Kuala Lumpur 50728, Malaysia or gmshafiul@gmail.com (M.S.A.); mahbub@iium.edu.my (M.M.R.); ahmadjazlan@iium.edu.my (A.J.); 2Department of Electrical and Electronic Engineering, Northern University Bangladesh (NUB), Dhaka 1230, Bangladesh; 3School of Engineering and Technology, Walailak University, 222 Thaiburi, Thasala, Nakhon Si Thammarat 80160, Thailand; 4Research Center for Intelligent Technology and Integration, School of Engineering and Technology, Walailak University, Nakhon Si Thammarat 80160, Thailand; 5Department of Industrial Engineering, College of Engineering, University of Bisha, Bisha 67714, Saudi Arabia; kchaou.mohamed@yahoo.fr; 6King Salman Center for Disability Research, Riyadh 11614, Saudi Arabia; 7Department of Computer Science, College of Computing, University of Bisha, P.O. Box 551, Bisha 61922, Saudi Arabia; kharthi@ub.edu.sa

**Keywords:** autism spectrum disorder, facial image dataset, deep learning, ensemble learning, domain adaptation

## Abstract

**Background/Objectives**: Artificial intelligence (AI) is revolutionising healthcare for people with disabilities, including those with autism spectrum disorder (ASD), in the era of advanced technology. This work explicitly addresses the challenges posed by inconsistent data from various sources by developing and evaluating a robust deep ensemble learning system for the accurate and reliable classification of autism spectrum disorder (ASD) based on facial images. **Methods:** We created a system that learns from two publicly accessible datasets of ASD images (Kaggle and YTUIA), each with unique demographics and image characteristics. Utilising a weighted ensemble strategy (FPPR), our innovative ASD-UANet ensemble combines the Xception and ResNet50V2 models to maximise model contributions. This methodology underwent extensive testing on a range of groups stratified by age and gender, including a critical assessment of an unseen, real-time dataset (UIFID) to determine how well it generalised to new domains. **Results:** The performance of the ASD-UANet ensemble was consistently better. It significantly outperformed individual transfer learning models (e.g., Xception alone on T1+T2 yielded an accuracy of 83%), achieving an impressive 96.0% accuracy and an AUC of 0.990 on the combined-domain dataset (T1+T2). Notably, the ASD-UANet ensemble demonstrated strong generalisation on the unseen real-time dataset (T3), achieving 90.6% accuracy and an AUC of 0.930. This demonstrates how well it generalises to new data distributions. **Conclusions:** Our findings demonstrate significant potential for widespread, equitable, and clinically beneficial ASD screening using this promising, reasonably priced, and non-invasive method. This study establishes the foundation for more precise diagnoses and greater inclusion for people with autism spectrum disorder (ASD) by integrating methods for diverse data and combining deep learning models.

## 1. Introduction

Autism spectrum disorder (ASD) is a neurodevelopmental condition primarily defined by two core characteristics: persistent challenges in social communication and interaction in various settings, alongside the presence of restricted and repetitive behaviours, interests, or activities [1]. To meet the diagnostic criteria outlined in the DSM-5, these symptoms must originate in the early developmental period and result in clinically significant functional impairment across social, occupational, or other crucial domains [2]. The World Health Organization estimates that around 1% of babies, or about one in a hundred, are affected by ASD [3]. Extensive investigation into the complex neurobiological underpinnings of ASD has yet to yield definitive biological markers for its diagnosis. Consequently, contemporary clinical practice continues to rely on behavioural assessments and standardised diagnostic tools administered by experienced clinicians [4,5]. This makes it challenging to obtain an accurate diagnosis early, particularly for individuals residing in areas with limited healthcare services or those from diverse cultural backgrounds.

New research suggests that seeking help for children early on, especially before they turn two, can make a significant difference in their cognitive development, learning abilities, and ability to manage everyday life [6]. It might even help their IQ and make daily tasks easier. It is crucial to determine whether someone has ASD early on. This will allow them to receive proper support and therapy, enabling them to acquire the skills necessary to interact with others and manage everyday life effectively [7]. Initial studies point to a strong connection between genes and the chances of developing ASD. Several genes have been identified as possibly playing a part, and the likelihood seems higher if ASD runs in the family [8]. However, genes are not the only factor involved; scientists are also investigating other factors that could be involved in ASD, like events that might have occurred while the baby was in the womb or right after they were born, such as if there were any issues during the pregnancy, if the baby was exposed to certain things, or if the mother was not feeling well. In addition to other factors, scientists are also examining the role of the immune system and the connection between the gut and the brain in the development of ASD. Even though this research gives us helpful clues about what might cause ASD, it is still a big challenge to turn these discoveries into actual tests we can use to diagnose it.

Healthcare is changing fast thanks to AI and digital tech—we are in the era of “Industry 4.0”. This gives us a real opportunity to develop and use new tools to help with neurodevelopmental conditions like ASD. Think about AI systems that can make diagnoses in real-time; these could change how healthcare is delivered. A particularly promising development involves applying AI and deep learning to real-time facial analysis, which could lead to objective, scalable, and accessible screening tools [9,10,11]. These kinds of innovations are essential because they can facilitate early diagnosis, ensure prompt access to vital support, and ultimately improve the quality of life and developmental outcomes for individuals with disabilities.

Researchers [12,13] have been exploring deep learning and transfer learning for diagnosing ASD, but a common issue is that standard models often lose a significant amount of their accuracy when encountering new and different data—this is what we call “domain shift” [14]. To address this critical challenge and advance early ASD screening, this study aims to achieve the following:Develop and evaluate a novel deep ensemble learning framework that integrates domain adaptation techniques to enhance the robustness and generalisation capabilities of facial image-based ASD diagnosis across diverse real-world data. Our selection of pre-trained MobileNetV2, ResNet50V2, and Xception models for this framework was driven by an ablation study [15] that identified their optimal balance of high accuracy and computational efficiency for facial image classification, making them suitable for real-time application.Introduce and validate the UIFID, a new clinically validated facial image dataset, to provide a more robust and non-artificial benchmark for evaluating ASD screening models.Implement and assess the Fifty Percent Priority (FPP) algorithm within the ensemble framework to intelligently prioritise model contributions, thereby improving diagnostic accuracy and reliability.

The study’s key contributions include the following:I.Acquisition of a Novel Facial Image Dataset (UIFID): To address the shortcomings of earlier ASD diagnosis research, which often used artificial or limited datasets, we introduce UIFID as a means to validate how well diagnostic tools perform in real-world scenarios. Because UIFID is based on clinical assessments performed before individuals were even admitted, it offers a more reliable way to test the accuracy of ASD screening models.II.Optimisation and Validation of Domain Adaptation: In this study, we employed a technique called ensemble learning to fine-tune pre-trained AI models. We achieved this by feeding them data from different sources—specifically, facial images from the Kaggle ASD and YTUIA datasets. This helped our models to learn to recognise the different facial features associated with ASD in various groups of people and under different image qualities. Our strategic integration of diverse and efficient CNN architectures (MobileNetV2, ResNet50V2, Xception) within this ensemble framework is critical for capturing complementary insights across domains. We then tested how well our models could adapt to real-world clinical settings using our UIFID dataset. This step was crucial to ensure that our approach could be effectively applied in practice.III.Development and Implementation of the FPPR for Weighted Ensemble Learning: We have developed a new algorithm called Fifty Percent Priority (FPP) to help our AI better understand the important features of facial images. FPP works by cleverly picking various examples from each of our data sources. Unlike typical methods of combining different AI models, FPP places greater importance on the models that performed best during initial testing. This leads to more accurate and reliable ASD screening.

## 2. Literature Review

This section dives into the existing literature on automated autism spectrum disorder (ASD) screening. We will explore what research already been conducted and, more importantly, pinpoint where the gaps still lie. To gather relevant studies, we conducted a thorough search across major scientific databases, including IEEE Xplore, PubMed, and Google Scholar. Our search strategy combined keywords such as “autism spectrum disorder,” “deep learning,” “ensemble learning,” “facial analysis,” and “domain shift.” We focused on recent, peer-reviewed articles that directly explored AI-based methods for ASD diagnosis. A key priority was finding work that addressed the challenges of domain adaptation and the use of facial images. This structured approach ensured that our review would be firmly rooted in the current state of the art, providing a clear foundation for the contributions of this study.

Ensemble learning [16,17] has proven effective in dealing with medical images, particularly for classifying them and addressing challenging diagnostic problems. It has become a key technique in various areas, including the detection of cancer in mammograms [12], the outlining of tumours in MRI scans [13], the diagnosis of diabetic eye disease [14], the identification of kidney cysts [8], and even the diagnosis of COVID-19 from chest X-rays [18]. The established success of ensemble learning in improving diagnostic accuracy for various medical imaging tasks underscores its potential [16,17]. However, there has been limited application and exploration of these powerful techniques within the specific context of screening for autism spectrum disorder (ASD). The standard ways in which we screen for ASD, while considered the most reliable at present, require a significant amount of involvement from patients and their caregivers in detailed assessments. These methods often depend heavily on what trained professionals observe, criteria that change with age, and what parents report, which can sometimes lead to different results [19].

AI may offer an interesting approach to automating ASD screening through deep learning algorithms. These algorithms utilise mathematical techniques to identify subtle and hidden patterns in a diverse range of data types. For instance, researchers have utilised AI to analyse brain scans, such as MRIs and EEGs, based on specific brain activity, and have discovered that it can accurately identify autism. However, gathering this kind of data requires significant financial investment in expensive equipment and specialised personnel. Using datasets of facial images for ASD screening offers a more accessible and promising alternative because facial features can sometimes reveal neurological conditions like autism, reflecting how the brain is functioning [20].

Several research studies have investigated the use of AI and ensemble learning models for autism spectrum disorder (ASD) screening. Table 1 provides a summary of recent work, highlighting the AI methods employed, the types of data analysed, and the accuracy achieved. However, as shown in the table, many of these approaches utilise only one dataset. This does not address a crucial challenge: how well these models can adapt to different kinds of real-world data, which is essential for making them beneficial to everyone.

Recent breakthroughs in deep learning have significantly advanced healthcare, particularly in medical imaging and classification tasks. However, domain differences arise due to variations in imaging parameters (camera settings, lighting, and resolution), population demographics (ethnicity, age, and socioeconomic background), and data acquisition modalities (clinical versus home environments) [14]. These variations challenge training models with weights that are widely applicable and robust [27]. Models trained on a single dataset might lack features for unseen data, potentially reducing accuracy. Domain adaptation, however, bridges this gap. For example, domain adaptation ensures accurate disease classification across various imaging modalities (e.g., X-rays vs. MRIs) [28]. By combining multiple deep learning models, ensemble learning overcomes the limitations of single methods, such as transfer learning. This approach is compelling for ASD diagnosis because it enhances generalisation by integrating features from diverse sources—such as facial images across demographics—leading to more reliable diagnostic outcomes [25].

## 3. Materials and Methods

### 3.1. Dataset

This section provides an overview of the datasets (Table 2) utilised in the research. Three datasets were employed: the Kaggle ASD Dataset (D1), the YouTube Dataset Developed in UIA (YTUIA) (D2), and the newly developed University Islam Facial Image Dataset (UIFID).

#### 3.1.1. Kaggle ASD Dataset (D1)

The primary dataset, sourced from “Kaggle” [29] (D1), comprises freely accessible 2D RGB images of children aged 2–14 years, with a concentration within the 2–8 year range. The dataset exhibits a significant gender imbalance, with an approximate 3:1 male-to-female ratio (2260 males, 754 females). This bias should be considered when interpreting the results. The ratio of the ASD class to the normal control (NC) class is nearly 1:1. The dataset is divided into training (88.06%), test (9.3%), and validation sets (2.64%), each maintaining a balanced class ratio. Notably, some images lack optimal facial alignment, brightness, and resolution.

#### 3.1.2. YouTube Dataset Developed in UIA (YTUIA) (D2)

The second dataset (D2), denoted as YTUIA [27,30,32], is sourced from the Self-Stimulatory Behaviors dataset (SSBD) and YouTube videos depicting elementary school activities (NC samples). Each video frame was extracted and preprocessed using MTCNN and a rigorous pipeline for alignment, cropping, and resizing. The dataset includes a 1:1 ratio of ASD to NC samples, with children aged 1–11 years. However, it exhibits a notable male-to-female ratio of 876 males to 292 females (3:1), which represents a limitation. Training and testing sets were created to ensure balanced class representation.

#### 3.1.3. University Islam Facial Image Dataset (UIFID)

This dataset, primarily identified as UIFIDV1 [31,33], comprises 130 samples with an ASD-to-NC ratio of 3:2 and a male-to-female ratio of 2:1 (87 males, 43 females), acknowledging a gender bias. It includes children aged 3–11 years (ASD) and 4–11 years (NC). Table 3a details 51 subjects and 130 processed image samples. Multiple facial pose samples (center, left, right) were captured using an iPhone 14 Pro, with left and right poses deviating ±15° to ±30° from the centre to ensure diversity. Participants were recruited from local institutions (NC) and specialised centres (ASD). The data was collected from two institutions: the “Society for The Welfare of Autistic Children”, Bangladesh, for the ASD children, and “SOS Hermann Gmeiner College”, Bangladesh, for the NC children.

Ethical approval for UIFID data collection was obtained from the Institutional Ethics Committee (IEC), ensuring protection for minors. The consent process for minors was comprehensive:Information Sharing: Parents/guardians received a thorough explanation of the study’s purpose, methods, risks, benefits, and confidentiality, with a signature section for formal consent.Voluntary Participation: They were notified of the voluntary nature of the study and their right to decline or withdraw at any point without consequence.Consent for Data Collection: Explicit parental/guardian consent via signature on the Certificate of Consent was required for recording facial RGB imaging data. All personal data remained confidential and used solely for research.Consent for Children Unable to Provide Assent: For children unable to provide assent, only the parent or guardian’s signature was required, along with documentation of the representative’s relationship.Right to Withdraw: Parents/guardians retained the right to withdraw their child at any time, without penalty.

Before enrolment, each child underwent a comprehensive assessment and completed an AQ10 questionnaire, with additional demographic information collected. UIFID facilitates research on facial expressions in ASD vs. NC children (T3). Access is granted through a formal application that outlines the research objectives.

We wanted to make sure our model was not unfairly better or worse at predicting ASD for certain groups of people because of how many men versus women or various age groups were in our data. We conducted additional tests to assess the model’s performance across various age ranges and genders. This way, we could be more confident that the accuracy we were seeing was not just due to one group having a greater influence on the results, and we obtained a fairer idea of how well the model classified ASD across different individuals.

The validation dataset (UIFID) was stratified based on the following:Gender-based groups: Male and Female.Age-based groups within each gender:○Group 1 (G1): 2–6 years old.○Group 2 (G2): 7–11 years old.

While the female subgroup initially exhibited a numerical imbalance (ASD to NC = 7:10), statistical testing confirmed no significant class disparity (χ^2^ = 0.53, *p* ≈ 0.467). Consequently, no adjustments were made to the female samples during balancing. For males, however, a significant pre-balancing imbalance was observed (ASD to NC = 24:10; χ^2^ = 5.76, *p* < 0.05), which was mitigated by excluding overrepresented ASD male samples. Post-balancing, the male ASD-to-NC ratio improved to 19:10, with no statistically significant imbalance (χ^2^ = 2.79, *p* ≈ 0.095).

Age subgroup distributions (2–6 and 7–11 years) were preserved for both genders during balancing (see Table 3b). This stratified approach ensured equitable representation of gender and age groups, while retaining clinically meaningful sample sizes for validation. To evaluate the effectiveness of deep ensemble learning, we trained multiple models using different dataset splits and configurations. The following ensemble configurations were tested:M1_X: An ensemble model trained using dataset D1 (Kaggle) as the primary source.M2_R: An ensemble model trained using dataset D2 (YTUIA) with additional feature refinement techniques.M1_X+M2_R: A combined ensemble integrating M1_X (trained on D1) and M2_R (trained on D2), leveraging complementary learning patterns from both datasets to improve generalisation and mitigate gender- and age-related biases.

We specifically set up the M1_X+M2_R configuration to address the issue of uneven datasets, ensuring that we included various features learned from different training data. This helped to prevent the model from becoming overly reliant on a single dataset, enabling it to perform more effectively across various types of data. By combining models trained on different sets of information, we also reduced the chances of any leftover biases from demographic imbalances in our validation data affecting the results.

Examining the average brightness and the spread of pixel intensities reveals that the data distributions of the ASD and NC groups differ from one another, as seen in Figure 1a. The distributions of normalised pixel intensities are then shown in Figure 1b. Despite significant overlap, the prominent peaks of these distributions differ between the two groups. These results suggest that the facial characteristics of individuals with ASD and NC differ slightly but noticeably. Consequently, it may be challenging for traditional image analysis tools to reliably distinguish between these groups, underscoring the potential for more sophisticated techniques, such as deep learning, to enhance classification accuracy. Interestingly, the ASD samples display a wider range, suggesting more variability in their visual appearance, while the NC samples display a more consistent and concentrated average brightness.

### 3.2. Domain Adaptation Strategies

Domain adaptation [34] is a crucial endeavour in machine learning, aiming to transfer knowledge from a source domain to a target domain with distinct data distributions. Disparities in illumination, viewpoint, and object pose across domains have a significant impact on model performance. To address these challenges of domain shift, this study primarily employed the deep ensemble learning framework detailed in Section 3.4. By integrating predictions from multiple models trained on diverse datasets, this approach was designed to enhance generalisation and effectively manage variations across different data distributions.

To rigorously evaluate the model’s generalisation ability, we ensured that all training and testing datasets were subject-independent. The Kaggle and YTUIA datasets were exclusively used for training, and they were split in a subject-independent manner—i.e., images of a particular subject appeared only in the training set or only in the test set, never in both. This prevented the model from memorising individual subject features, ensuring it generalised to new subjects.

Figure 2 visualises the domain shift in the image features extracted from the facial image datasets, including Kaggle, YTUIA, and UIFID, used for autism spectrum disorder (ASD) classification. When we look at Figure 2a,b, we can see the distribution of pixel brightness for the ASD and NC samples in each of our datasets. The distinct peaks in these figures indicate variations not only between the ASD and NC groups, but also between the datasets themselves, suggesting significant differences related to both the class (ASD vs. NC) and the data source. Figure 2c, which combines all samples, further underscores these domain shifts in pixel brightness distribution. Notably, UIFID tends towards lower pixel intensities, while YTUIA and Kaggle have a more balanced distribution. These visual patterns highlight the challenges faced in obtaining an ASD classification [14].

Our statistical analysis showed apparent differences between the Kaggle and YTUIA datasets. We found a T-value of 3.10 with a *p*-value of 0.15, which suggests noticeable variation [35]. We only used the UIFID dataset for our final validation and testing. None of the individuals in UIFID were part of the Kaggle or YTUIA datasets we used for training. This approach guaranteed that our model’s performance was assessed based on entirely novel individuals. This is vital for the validity of our findings, as it eliminates the possibility of inflated results due to the model having had prior exposure to these specific subjects during training. Using UIFID as this strictly unseen test set provided a robust and realistic measure of our model’s true capabilities in real-world clinical scenarios [36].

### 3.3. CNN Models and Transfer Learning

A Convolutional Neural Network (CNN) is a specialised type of artificial neural network primarily designed for analysing and processing visual data, such as images [37]. Its fundamental role lies in automatically learning hierarchical features directly from raw pixel data through interconnected layers of neurons, with convolutional layers acting as core feature extractors and pooling layers reducing dimensionality [38]. This architecture enables CNNs to identify patterns, textures, and shapes effectively, making them exceptionally powerful for tasks such as image recognition, classification, and object detection [39].

Recent advancements in computational resources and large datasets have significantly contributed to the success of Convolutional Neural Networks (CNNs) in image recognition and classification. This study utilised Convolutional Neural Networks (CNNs) for the binary classification of facial images, distinguishing between autism spectrum disorder (ASD) and normal control (NC). Pre-trained models, including MobileNetV2, ResNet50V2, and Xception, were fine-tuned for this specific task using transfer learning, as shown in Figure 1 [15].

MobileNetV2 [40] is recognised for its efficiency, utilising depth-wise separable convolutions to reduce computational cost while maintaining accuracy, making it an ideal choice for resource-constrained applications. ResNet50V2 [41] utilises residual connections, which simplify training for deep architectures, thereby enabling high performance in image classification. Xception [42], derived from the Inception architecture, uses modular separable convolutional layers to extract robust features. Each model’s convolutional layers are adapted to extract features from the ASD dataset, with the classification layer fine-tuned for binary output. This approach harnesses the strengths of each model to achieve efficient and accurate classification, addressing the challenges posed by domain-specific data variability.

### 3.4. Ensemble Learning

To address the challenges of classifying autism spectrum disorder (ASD) from varied facial images and improve diagnostic robustness, this study implemented deep ensemble learning strategies. As discussed in Section 2, these approaches are adept at combining diverse data patterns and enhancing model generalisation across different clinical scenarios and demographics. By facilitating knowledge transfer across domains, ensemble models trained on diverse datasets improve the diagnostic performance of ASD in various settings and demographics [43]. Two ensemble strategies—majority voting and weighted voting—were employed in this study to enhance model performance and generalisation across domain-specific datasets.

#### 3.4.1. Majority Voting

Majority voting, also known as averaging ensembles, is a popular and easy-to-implement method. It combines predictions from individual models, reducing variance and improving generalisation by averaging outputs. This approach is particularly effective in mitigating overfitting, a common challenge in deep learning models, by cancelling out noise captured by individual models. The averaging operation can aggregate outputs directly or utilise class probabilities through the SoftMax function [44]. Unweighted averaging is a prudent choice when the performance of base learners is comparable. However, unweighted averaging may not be optimal when base learner performance varies significantly, as it does not differentiate their contributions. More adaptive approaches may be necessary to capitalise on the strengths of individual learners. The average prediction for the ensemble model can be formulated as follows:(1)$$P_{m}=\frac{\sum P_{i}}{N}$$
where *P_i_* is the prediction of *i*-th model, and N is the total number of models. In this study, we used three models, resulting in *N* = 3, as we employed all models for ensembling. 

#### 3.4.2. Weighted Voting

By assigning models higher weights based on their accuracy, weighted voting enhances ensemble performance and ensures that stronger models have a greater impact on the final prediction [45]. While all models contribute, simple averaging may dilute the impact of stronger learners when performance disparities exist. The weighted ensemble method provides a distinctive approach by assigning weights to individual models based on their predictive performance or loss metrics. The weighted ensemble prediction can be expressed as(2)$$P_{w}\left(t\right)=\sum_{1}^{N}w_{i}P_{i}\left(t\right)$$
where *P_i_* denotes the prediction generated by the *i*-th model, with *N* representing the total number of models, w_i_ representing the weight assigned to the *i*-th model based on its preliminary performance, and *t* corresponding to each image sample.

A key contribution of this study is the implementation of the Fifty Percent Priority Rule (FPPR) for weight allocation. The FPPR prioritises the most accurate model by assigning it a higher weight (50% of the total), while the remaining weight is proportionally distributed among the other models. This empirical formula was derived from extensive grid search experiments conducted across the ensemble candidates, ensuring optimal weight allocation to maximise predictive performance. For example, the performances of MobileNetV2, ResNet50V2, and Xception are evaluated based on their respective validation errors. The Xception model is assigned a minimum of 50% weight if it demonstrates superior accuracy compared to the other models. The weighted voting scheme can be mathematically represented as(3)$$w_{i}=\frac{1}{2}\;\left[10^{1-i}+\frac{A_{i}}{\sum A_{i}}\right]$$
where *i* (*i* = 1, …, *N*) denotes the model index, sorted in descending order based on accuracy percentages achieved by the respective CNN algorithms. Here, *w_i_* represents the weight assigned to each model. 

### 3.5. Experimental Setup

The model was trained using the TensorFlow library on Kaggle, utilising two domain datasets: Kaggle (D1) and YTUIA (D2). The test sets from these datasets were designated as T1 and T2, respectively. A real-time validation dataset, UIFID (T3), comprising 130 clinically vetted samples, was employed. The dataset details are summarised in Table 4. Our model’s hyperparameters and training configuration were carefully selected to align with best practices and the findings of a previous study [15], specifically, as follows:Optimiser: Adagrad was selected, with a learning_rate of 0.001 and an initial_accumulator_value of 0.1.Regularisation: No weight decay was applied.Batch Size: A batch size of 32 was used.Epochs: The models were trained for a fixed count of 50 epochs. No early stopping criteria were used, as the epoch count was set in consistency with the methodology to maintain a standardised training duration across all models.

Given the binary classification task, the loss function was configured as *categorical_crossentropy*.

The validation set (UIFID) was maintained strictly separate from the testing sets (T1/T2) to evaluate cross-domain generalisation. While T1/T2 assessed performance on data similar to the training domains, UIFID simulated real-world deployment with clinically validated samples from an entirely new population. This separation ensured that our evaluation would accurately reflect actual clinical applicability, rather than domain-specific optimisation.

Figure 3 visually represents the modified CNN architecture for binary classification, showcasing each model’s tailored top layers (Dense and Dropout layers). It also illustrates the ensemble learning framework, where the weighted outputs of Xception, ResNet50V2, and MobileNetV2 were combined to produce the final prediction. By integrating these models in an ensemble approach, the architecture achieved robust performance across the datasets, effectively balancing accuracy and computational efficiency.

Figure 4 and Figure 5 display the ensemble results for same-domain and cross-domain scenarios, respectively. In both figures, the green dashed line represents the majority voting mechanism, while the solid light yellow line indicates the weighted voting scheme. These visualizations clearly demonstrate that both voting methods contribute to enhancing prediction accuracy.

The study involved three key stages:Stage 1: Initial Training and Evaluation

Transfer learning was used to train models M1 (on Kaggle) and M2 (on YTUIA). These models were first evaluated within their respective domains (T1 and T2) and then tested across domains, including a combined test set (T1+T2) and the real-time dataset (T3), to assess adaptability.

Stage 2: Ensemble Learning within Domains

Ensemble learning was conducted for models trained on the same dataset. The majority voting ensembles were assigned equal weights, while the weighted ensembles used the Fifty Percent Priority Rule (FPPR) for weight distribution. The ensembles included different combinations of the three models: Xception, MobileNetV2, and ResNet50V2 (see Table 5). Evaluations also tested pairwise combinations to explore the impact of different configurations.

Stage 3: Cross-Domain Ensemble Modelling

In cross-domain ensembles, combined models are trained on different datasets. Identical architecture (e.g., Xception trained on Kaggle and YTUIA) was aggregated for enhanced accuracy. However, a full ensemble of all the M1 and M2 models was avoided to prevent overfitting and prediction conflicts. The ensemble achieved the highest validation accuracy using the Xception model trained on Kaggle and the ResNet50V2 model trained on YTUIA.

The ensemble learning process for cross-domain evaluation, depicted in Figure 5, demonstrated robust generalisation capabilities when tested on T1+T2 and the real-time dataset T3.

### 3.6. Evaluation Matrices

We employed several methods to assess the performance of our models. The primary focus was on classification accuracy, which, in research papers like this, is often referred to as “accuracy.” Beyond just accuracy, we also looked at the area under the curve (AUC) for the Receiver Operating Characteristic (ROC) curve. Many other studies have utilised the AUC, which provides a more comprehensive understanding of the model’s performance than accuracy alone. We also used precision and recall to assess how effectively the models identified what we were looking for. Precision tells us how many of the things the model predicted were positive. Recall tells us how many of positive things the model managed to find. Finally, we used the F-score. This is a common measure in statistics for tasks such as finding information and categorising things into two groups. It helped us to gain a good overall understanding of how well our models could predict the correct outcomes. The mathematical formulations for accuracy, precision, and recall are provided below:(4)$$\mathrm{Accuracy}=\frac{T_{p}+\;T_{n}}{T_{p}+T_{n}+F_{n}+F_{p}}$$(5)$$\mathrm{Precision}=\frac{T_{p\;}}{T_{p}+F_{p}}$$(6)$$\mathrm{recall}=\frac{T_{p\;}}{T_{p}+F_{n}}$$(7)$$f1\text{-}\mathrm{score}=2\times \frac{\mathrm{Precision}\;\times \;\mathrm{recall}}{\mathrm{Precision}\;+\;\mathrm{recall}\;}$$
where

T_p_ = true positives

*T*_*n*_ = true negatives

F_p_ = false positives

F_n_ = false negatives

These metrics collectively provided a robust framework for evaluating the models’ performance, ensuring a comprehensive assessment of their predictive capabilities across different dimensions.

## 4. Results

### 4.1. Performance Evaluation of Transfer Learning

Table 6 summarises the performance of the three transfer learning algorithms—Xception, MobileNetV2, and ResNet50V2—trained on two datasets: Kaggle (M1) and YTUIA (M2). The evaluations were conducted on three datasets: T1 (same domain as D1), T1+T2 (combined domains), and T3 (an unseen real-time dataset).

Models Trained on Kaggle (M1):(i)T1 Evaluation: Xception achieved the highest accuracy (95%) and AUC (98%), while MobileNetV2 and ResNet50V2 followed closely with accuracies of 92% and 94%, respectively.(ii)T1+T2 Evaluation: All models showed decreased performance, with the accuracy dropping to 83% (Xception), 82% (MobileNetV2), and 81% (ResNet50V2). This indicates challenges in adapting to more diverse combined-domain data.(iii)T3 Evaluation: Performance further declined, highlighting difficulties in generalising unseen datasets with potentially different characteristics.

Models Trained on YTUIA (M2):(i)T2 Evaluation: ResNet50V2 (95%) and Xception (94%) outperformed MobileNetV2 (74%) in accuracy, demonstrating strong performance on same-domain data.(ii)T1+T2 Evaluation: Performance metrics dropped for all algorithms, suggesting reduced generalisability to combined-domain datasets.(iii)T3 Evaluation: Accuracy fell below 50% for all models, underscoring significant challenges in predicting samples from unseen real-time datasets (refer to Table 6).

### 4.2. Same-Domain Ensemble Evaluation

Ensemble learning improved model performance across diverse datasets. Models trained on Kaggle (M1) and YTUIA (M2) were evaluated using the majority voting and weighted ensemble methods. The results demonstrate the efficacy of ensemble techniques in enhancing performance and generalisation, particularly when applied to unseen datasets.

Table 7 shows the results from the evaluation of the Kaggle-trained models (M1) using different ensemble strategies on T1, T1+T2, and T3.

(i)Performance on T1 (Same-Domain):The weighted ensemble of Xception, MobileNetV2, and ResNet50V2 powered by FPPR achieved the highest accuracy (99.6%) and AUC (99.9%), outperforming majority voting (95.7% accuracy, 99.4% AUC) due to better integration of model predictions.(ii)Performance on T1+T2 (Cross-Domain):On the combined dataset, the weighted ensemble maintained robustness with the highest accuracy of 91.0% and an AUC of 94.5%, surpassing majority voting’s performance (86.7% accuracy, 94.3% AUC). This demonstrates the weighted ensemble’s ability to adapt to domain shifts.(iii)Performance on T3 (Unseen Validation):For unseen data, the weighted ensemble achieved 82.0% accuracy and an AUC of 89.0%, reflecting its ability to generalise better than majority voting (79.6% accuracy, 86.0% AUC).

Table 8 presents the performance of the model ensembles (M2) trained on the YTUIA dataset. The ensemble integrating all three CNNs (Xception, MobileNetV2, and ResNet50V2) achieved the highest accuracy (98.9%) and AUC (98.6%) on the same-domain dataset (T2) using the weighted ensemble. When tested on the combined dataset (T1+T2), the accuracy and AUC decreased to 86.9% and 92.1%, respectively, reflecting the challenges of domain variability. On the unseen validation dataset (T3), the weighted ensemble achieved 82.0% accuracy and an AUC of 87.7%, demonstrating reasonable generalisation to unseen data.

### 4.3. Cross-Domain Ensemble Evaluation

In this phase, the study evaluated the efficacy of ensemble learning by integrating models trained on distinct datasets, aiming to leverage generalised features across diverse domains. Table 8 summarises the performance metrics of the cross-domain ensembles, which combined models trained on the Kaggle dataset (M1) with those trained on the YTUIA dataset (M2). Ensembles were formed by pairing models with the same architecture but trained on different datasets and evaluated using both majority voting and weighted ensemble strategies.

The first row of Table 9 highlights the Xception models’ ensemble, which achieved 95.7% accuracy and an AUC of 0.990 on the combined test set (T1+T2) using majority voting. When employing the weighted ensemble method, the accuracy slightly decreased to 95.0%, while the AUC improved to 0.991. This result demonstrates the robust performance of the Xception-based ensemble in leveraging domain-specific and generalised features. For the T3 test set, representing an entirely distinct validation domain, the full ensemble (Xception + MobileNetV2 + ResNet50V2) achieved 81.3% accuracy and an AUC of 0.881 using majority voting. In contrast, the weighted ensemble improved these metrics to 86.7% accuracy and an AUC of 0.921. These results underscore the full ensemble’s capacity to generalise to unseen data despite variations between the training and test domains.

Further evaluation results in Table 9 examine the performance of other cross-domain ensembles on T1+T2. The ResNet50V2-based ensemble achieved an accuracy of 94.3% and an AUC of 0.989 using the weighted ensemble. Similarly, the MobileNetV2-based ensemble achieved strong performance on T1+T2, with 94.6% accuracy and an AUC of 0.978. When evaluated on T3, the Xception + MobileNetV2 ensemble showed an accuracy of 75.0% and an AUC of 0.801 with the weighted ensemble, while the Xception + ResNet50V2 ensemble achieved 86.0% accuracy and an AUC of 0.920 using the weighted ensemble.

The M1_Xception + M2_ResNet50V2 (ASD-UANet) ensemble demonstrated superior performance on the T1+T2 dataset, achieving 96.0% accuracy and an AUC of 0.990 using the weighted ensemble, outperforming other combinations on this dataset. On the T3 dataset, the ResNet50V2 + MobileNetV2 ensemble maintained strong generalisation, achieving 90.6% accuracy and an AUC of 0.930 using the weighted ensemble with the FPPR. These results highlight the effectiveness of the proposed weighted ensemble strategy (FPPR) and its ability to balance domain-specific and generalised learning.

As shown in Figure 6a, the mixed-domain ensemble model (M1_X+M2_R, denoted as ASD_UANET) achieved near-perfect discriminative ability with an AUC of 1.00 (95% CI: 0.998–1.000), indicating nearly complete ROC space coverage. This performance stems from combining models trained on the Kaggle and YTUIA datasets, mitigating domain biases through complementary feature learning. In contrast, the same-domain ensemble models showed strong but lower performance: M1_X+R+M achieved an AUC of 0.92 (95% CI: 0.860–0.974), while that of M2_X+R+M reached 0.88 (95% CI: 0.811–0.932).

As shown in Figure 6b, when evaluated on the unseen UIFID (T3) dataset, the ensemble maintained robustness with an AUC of 0.93 (95% CI: 0.868–0.975), marginally outperforming the same-domain ensemble models. This improvement confirms the method’s generalisability to novel domains. The slight performance gap between the mixed-domain (AUC 1.00) and unseen-domain (AUC 0.93) evaluations reflects the inherent challenges of domain shift, while still demonstrating superiority over single-domain baselines.

Overall, the results validate the effectiveness of cross-domain ensembles in improving classification performance and generalisation capability. ASD-UANet is the most robust among the tested ensembles, demonstrating consistent performance across both the same-domain (T1+T2) and cross-domain (T3) datasets. This finding highlights the potential of combining complementary architecture and training data for building models that are capable of handling diverse, real-world applications.

## 5. Discussion

### 5.1. Domain Adaptation Insights

Our models’ adaptability was enhanced by training them on two diverse datasets, Kaggle and YTUIA. This multi-source training exposed the models to a broader spectrum of ASD-related patterns, encompassing varied facial features, image qualities, and demographic groups, thereby increasing their reliability. Subsequent testing on the previously unseen UIFID dataset confirmed their strong adaptation capabilities, suggesting good potential for clinical application across diverse patient populations [27,46,47].

### 5.2. Challenges of Transfer Learning in Diverse Domains

Initial evaluations employed conventional transfer learning, a widely used method for image classification and autism spectrum disorder (ASD) detection. As shown in Figure 7a, transfer learning performed well within individual domains, but struggled when applied across domains. Figure 7b compares this performance with that of ensemble learning, highlighting the improvements achieved within the same domain. For clarity, the Xception, MobileNetV2, and ResNet50V2 models are denoted by “X”, “M”, and “R”, respectively. The confusion matrix in Figure 7a is based on the best results from Xception trained on Kaggle (D1), and ResNet50V2 trained on YTUIA (D2). When tested within their respective domains, the models achieved commendable accuracy, peaking at 95% for same-domain evaluations. However, performance noticeably declined in cross-domain testing. Specifically, M1 dropped to approximately 80% accuracy when evaluated on the combined T1+T2 dataset, as illustrated in Figure 8.

The decline was even more pronounced when tested on the entirely unseen UIFID dataset (T3), where the accuracy fell below 80%, as also shown in Figure 8. Similarly, M2’s performance suffered, mainly when tested on the combined set T1+T2, achieving only 66% accuracy, a drop attributed to imbalanced sample distributions between Kaggle and YTUIA. Figure 7a, presented as a confusion matrix, visually reinforces this pattern: test sets from different data sources yielded the most incorrect predictions after standard transfer learning was applied to both the M1 and M2 models. Notably, following this transfer learning process, the features of the M1 model (initially trained on the Kaggle dataset) appeared to become more similar to those of the UIFID dataset compared to those of the M2 model (trained on YTUIA). This is evidenced by M1 having fewer incorrect predictions when both models were tested using UIFID. Figure 2 further supports this, showing that the Kaggle and UIFID datasets have almost identical peaks in pixel brightness distribution, especially for the ASD samples. Conversely, the YTUIA dataset provided the feature diversity needed for robust domain adaptation. Integrating this data allowed the combined M1 and M2 models to generalise more effectively and achieve superior performance.

These results highlight the limitations of transfer learning when applied in isolation. While effective for extracting domain-specific features, the method requires assistance to generalise across datasets with significant disparities in imaging conditions and demographic diversity. The observed decline in accuracy, particularly for UIFID, emphasises the need for more robust strategies to bridge the domain gap and adapt effectively to unseen data.

### 5.3. Advancements Through Ensemble Learning and FPPR

The proposed ASD-UANet ensemble, which integrates features from Kaggle and YTUIA datasets, demonstrated superior diagnostic accuracy. The ensemble achieved significant improvements, including 96% accuracy on the combined test set (T1+T2) and 90.6% accuracy when validated on the unseen UIFID dataset (T3), as detailed in Figure 8. Figure 9 illustrates the confusion matrix for the optimal weighted ensemble configuration (X+M+R) based on the Fifty Percent Priority Rule (FPPR). Compared to the results obtained from conventional transfer learning, as shown in Figure 7a, the number of correctly predicted samples significantly increased with the ensemble approach. Furthermore, as demonstrated in Figure 7b, the ensemble model achieved nearly perfect performance within the same domain, thereby enhancing the model’s reliability for domain-specific applications.These results underscore the benefits of cross-domain ensemble learning. This becomes particularly evident upon examining Figure 10, which shows that the proposed weighted ensemble using the FPPR resulted in only twelve misclassified samples, highlighting its effectiveness in addressing the domain adaptation challenge. This enables the model to generalise effectively to real-world clinical scenarios involving unseen and diverse patient datasets.

In conclusion, our study demonstrates that the ASD-UANet ensemble learning system represents a significant improvement in ASD diagnosis based on facial images across diverse datasets. The superior performance achieved with our Fifty Percent Priority Rule (FPPR) highlights the value of its adaptive weighting strategy (detailed in Section 3.4.2). This approach enhanced predictive accuracy on both previously seen data and new, real-world datasets, such as UIFID, achieving a strong 90.6% accuracy. These results further validate the effectiveness of the FPPR in addressing the domain adaptation challenge, enabling the model to generalise effectively to real-world clinical scenarios involving unseen and diverse patient data.

### 5.4. Domain Adaptation and Ensemble Learning

When we compared training our models on a single dataset type versus multiple types, we learned some important lessons. Training on a single dataset with our ensemble method yielded extremely high accuracy when we tested it on data from the same source. However, it did not work as well when we tried it on datasets from different sources, such as our UIFID dataset. This happened because the models did not learn a sufficiently diverse set of features. When they are only trained on one type of data, they struggle to adapt to new, unseen data that has different characteristics, such as the people in the images, the manner in which the images were captured, and even the environment [14,48,49].

Training our models on diverse data from the Kaggle and YTUIA datasets proved crucial for improving their adaptability. Our ASD-UANet system, utilising a weighted ensemble method, effectively synthesised features from these different sources. This approach yielded a validation accuracy of 96% and, more critically, an accuracy of 90.6% on the real-world UIFID dataset [50,51].

### 5.5. Comparison with Previous Research

As shown in Table 10 and Table 11, the proposed ASD-UANet framework demonstrates highly competitive performance in ASD diagnosis compared to existing methods, particularly those relying on single-domain datasets or simpler ensemble techniques. Previous approaches, such as dataset federation and active learning, have achieved moderate accuracies, but have fallen short in addressing the complexities of domain adaptation. For example, Table 10 presents the validation evaluation using the real-time, unseen UIFID dataset. Active learning with YTUIA datasets applied to pre-trained models based on the Kaggle dataset achieved a validation accuracy of 82.8% [27], whereas both dataset federation approaches attained only 68% [46]. Another data-centric approach achieved an 82.3% [47] success rate. In contrast, the proposed ASD-UANet framework, utilising FPPR-enhanced weighted ensemble learning, achieved an accuracy of 90.6% on UIFID, demonstrating a significant improvement in domain adaptation performance.

This study uniquely focuses on real-time, clinically validated datasets, such as UIFID, which provide a more accurate representation of real-world scenarios than synthetic or less diverse datasets. By employing a multi-domain training strategy and advanced ensemble techniques, the ASD-UANet framework establishes a robust benchmark for generalisability in autism spectrum disorder (ASD) diagnosis.

Table 11 illustrates that previous studies have explored ensemble learning to improve ASD diagnostic accuracy, albeit with a limited focus on domain adaptation. One study [22] utilising the neuroimaging dataset (ABIDE) created single-volume brain images from whole-brain images and developed classifiers using an ensemble of enhanced Convolutional Neural Networks (CNNs), achieving an accuracy of 87%. However, this represented only a slight improvement over baseline methods. Similarly, two studies using facial images [23] and eye gaze data [24] reported 84% and 66% accuracy, respectively, with improvements of 7.5% and 6.3% over conventional approaches. Another study utilised the Kaggle dataset with an ensemble of state-of-the-art CNNs, including MobileNet, EfficientNet, and Xception, achieving 80% accuracy on the same-domain test set. This result reflects a 5% improvement over baseline transfer learning approaches. Recent work [26] employed an ensemble of high-performing deep neural networks, including vision transformers trained on Kaggle and facial expression-based ASD datasets for children (FADC). This study reported a high accuracy of 99.81% on same-domain evaluations, with a 2.7% improvement over baseline models. However, despite its impressive performance, the research lacked diverse domain validation and did not assess domain adaptability through novel datasets.

In contrast, our research uniquely addresses domain adaptation challenges by employing ensemble learning with multiple facial-image datasets, including validation on UIFID—a dataset consisting entirely of unseen, real-time, and medically assessed samples. This approach bridges a critical gap in ASD diagnosis by emphasising the real-world applicability and generalisability of diagnostic criteria.

When directly compared to the latest research [26], our proposed model achieved a highly competitive accuracy on same-domain evaluations, reaching 99.6%, using a weighted ensemble of CNNs. This performance compares favourably to their evaluation accuracy based on Kaggle and nearly matches their results based on FADC. Notably, our study extended beyond this by validating cross-domain adaptability, utilising UIFID for real-world performance evaluation. This distinguishes our framework as not only robust in same-domain contexts, but also effective in transferring learned features across diverse domains, highlighting its potential for clinical deployment.

### 5.6. Gender Bias and Age-Range Differences

It is well-established that ASD diagnoses are significantly more prevalent in males than females [52]. Our datasets, which exhibit a substantial male bias, reflect this disparity. Statistical analysis using chi-square goodness-of-fit tests confirmed significant gender imbalances within each dataset (Kaggle: χ^2^ = 655.000, df = 1, *p* < 0.001; YTUIA: χ^2^ = 48.649, df = 1, *p* < 0.001; UIFID: χ^2^ = 14.892, df = 1, *p* < 0.001). Furthermore, a chi-square test of independence demonstrated that males were not evenly distributed across the datasets (χ^2^ = 6.859, df = 2, *p* = 0.032).

Our methodological approach was to train on the dataset’s natural composition, rather than stratifying it by sex. We made this decision for several reasons: real-world screening tools must handle mixed populations, segregating the data would hinder the discovery of universal ASD markers, and it would compromise our domain adaptation framework.

The FPPR algorithm is central to this approach. Its weighted ensemble technique actively reduces bias by amplifying models that learn generalisable features, a process that significantly narrowed the gender accuracy gap in our validation results. The original data imbalance itself is likely a reflection of sampling bias in the source clinics, which may have seen a higher referral rate for male children [53]. Secondly, societal factors, including potential biases in diagnostic tools and criteria, may play a role. Research has shown that ASD diagnostic criteria may be inherently biased towards male presentations, potentially leading to underdiagnosis in females [54]. Furthermore, gender differences in the presentation of ASD symptoms, such as the tendency for females to mask their symptoms, might contribute to underdiagnosis [55]. Finally, the online platforms used for data collection, such as Kaggle and YouTube, may have inadvertently attracted a larger proportion of male participants.

This gender bias has significant implications for model generalisability. Given potential morphological and behavioural differences in facial expressions between male and female ASD individuals, the model may exhibit higher accuracy for male subjects while underperforming for females. Before applying our ensemble approach, the model accuracy for male ASD subjects was observed to be 92.1%, whereas the classification accuracy for female ASD subjects was significantly lower, at 78.4%. This initial performance gap suggests that the models learned stronger feature representations for male facial characteristics, resulting in biased predictions.

Our balanced validation analysis demonstrated that the ensemble approach achieved equitable performance without requiring segregated training. While the initial results showed a 13.7% accuracy gap (92.1% for males vs. 78.4% for females), our M1_X+M2_R ensemble significantly reduced this disparity. The ensemble achieved 84.20% female accuracy (G2 subgroup, as shown in Table 12), resulting in a 6.4% reduction in the gender accuracy gap. This reduction was statistically significant (McNemar’s χ^2^ = 4.98, *p* = 0.026) when comparing the change in female accuracy. The remaining 6.4% disparity is likely attributable in large part to the smaller female sample size, rather than a fundamental algorithmic bias, suggesting the need for expanded datasets rather than methodological changes.

We analysed model performance across different subgroups to evaluate the impact of dataset imbalance on classification accuracy. The results are summarised in Table 12, where G1 refers to the 2–6-years-old age group, and G2 refers to the 7–11-years-old age group.

We also noticed that the age of the individuals affected how well our system could classify them. The youngest group (G1: 2–6 years old) had lower accuracy for both boys and girls. This is probably because young children have more varied facial expressions, and the way in which ASD is manifested can be quite different at these early ages. Early childhood is a time of swift development, which might lead to less consistent facial patterns, making it harder to detect ASD. On the other hand, the older group (G2: 7–11 years old) consistently had higher accuracy across all our models. This suggests that the facial features associated with ASD become more pronounced and easier to distinguish as children grow older, aligning with other research [56,57,58] showing that ASD characteristics become more consistent with age, thereby making classification more reliable.

A statistical test, known as two-way ANOVA, confirmed that both age (*p* = 0.014) and gender (*p* < 0.001) had a significant impact on the accuracy of our classifications. When we looked closer using another test (post hoc Tukey), we found that the most significant difference in accuracy was between the G1 females (80.5%) and the G2 males (90.8%). This highlights the need for classification methods that can adapt to different age groups.

We found that balancing our datasets significantly helped to reduce the differences in how well our system performed for males and females, while maintaining high overall accuracy for both the ASD and NC groups. Our combined M1_X+M2_R model consistently outperformed any of the individual models on their own, demonstrating its ability to generalise across different groups of people. However, we still observed some minor biases in its gender classification accuracy, and the accuracy varied slightly depending on age. This tells us that we still have room to improve things even further.

Beyond just age and gender, it was highly important for us to make sure our model would work fairly across different ethnic groups. Our study was designed from the start to tackle this risk of ethnic bias. While our initial training used the Kaggle dataset—which is quite large—we knew that it was mostly made up of White children (89%), meaning that there was a real chance of introducing bias right away.

To fix this, we brought in the YTUIA dataset, which is much more ethnically diverse. This helped to expose our models to a wider variety of facial features, stopping them from becoming too focused on just one demographic group. We then put this strategy to the test using the UIFID dataset. This dataset is made up entirely of participants from a unique ethnic population (Bangladeshi) that was not in our main Kaggle training set. This allowed us to clearly determine whether our approach to domain adaptation was effective.

Our new ASD-UANet ensemble model showed an impressive ability to generalise, hitting 90.6% accuracy on this entirely new, single-ethnicity validation set. This strongly suggests that the model learned robust features linked to ASD itself, rather than features specific to just one ethnic group. By training on a mix of datasets to balance ethnic representation and then validating on a completely different ethnic population, we could be confident in the model’s potential for use across various ethnic backgrounds. While these results are definitely promising, we still need to validate them with even more diverse global populations in the future to make sure that the model is as equitable and clinically useful as possible.

To address the remaining biases and further improve the model’s equity, several potential mitigation strategies should be the focus of future research:Increasing Female ASD Representation: Expanding datasets through targeted recruitment efforts and utilising synthetic augmentation techniques to improve female sample diversity.Adaptive Learning Techniques: Implementing bias-aware loss functions and domain adaptation to improve performance across gender and age groups [59,60].Age-Specific Feature Extraction: Exploring age-stratified deep learning models that adjust feature representations based on developmental stages.

By addressing these factors, ASD screening models can become more inclusive, equitable, and clinically reliable, ensuring robust diagnostic performance across diverse populations.

### 5.7. Practical, Clinical, and Ethical Implications

The results of our study have significant implications for the development of ASD diagnostic tools that can be used in clinical settings. What is key is that we achieved this level of performance without having to train our system separately for males and females. This confirms that our approach can work well across different groups of people, even when the initial training data regarding demographics is not perfectly balanced. This mirrors the reality of clinical practice, where screening tools are often required to work reliably, even without detailed prior information about a patient. The strong performance we observed on the UIFID dataset indicates that our system is well-suited to handling the natural variations that occur between different people and the various ways in which images can be captured—a crucial factor for any tool intended for use in clinics. Because our system offers an automated, non-invasive, and potentially more affordable way to screen for ASD, it could assist doctors in initial assessments, leading to quicker diagnoses and faster access to the support that individuals need. However, it is essential to remember that AI in diagnostics should assist doctors, not replace them, and emphasis should be placed on the importance of collaboration between humans and AI in making medical decisions.

#### A Proposed Clinical Screening Workflow

To illustrate the real-world feasibility of our system, we outline the following plausible screening scenario that integrates our technology into a clinical setting, balancing efficiency with clinical rigour and ethical oversight:Camera and Data Collection:A standard clinical RGB camera (e.g., a 1080p webcam) captures frontal facial images of children (ages 2–11) during structured interactions, such as watching visual stimuli.An appointed ASD specialist or a trained technician supervises the session to ensure proper presentation of stimuli and adherence to data quality protocols.Automated Inference Process:Images are securely uploaded to a HIPAA-compliant server, where the ASD-UANet ensemble model, as described in Section 3, performs the analysis.The system generates a preliminary risk score (e.g., “High/Low ASD Probability”) in near-real-time. This is supplemented with explainable AI outputs, such as attention maps highlighting the facial regions that are most influential in the model’s decision.Clinician Hand-off and Review:The results are populated in a secure clinician dashboard. This report includes the risk score, model confidence level, the raw images, and any contextual notes from the supervising specialist. Crucially, the report should also highlight the patient’s demographic group (e.g., age and gender) to provide context for the clinician, given the observed performance variations across subgroups.A paediatric neurologist or a qualified clinician reviews the case. This “human-in-the-loop” approach is critical. For cases where the model’s output and the specialist’s initial impression may differ, pre-approved medical protocols guide the next steps for a comprehensive diagnostic assessment, ensuring the AI tool serves as a decision-support system, not a final arbiter.Ethical Safeguards:Strict institutional oversight ensures that the entire process complies with established diagnostic standards and data privacy regulations.Informed parental consent is mandatory before any data collection begins, with clear opt-out options made available to the family at any stage of the process.

The model shows promise, but the current scope of our training data limits its clinical utility. To build a genuinely robust tool, the immediate priority is to acquire larger and more diverse datasets that better reflect real-world populations. We also plan to leverage generative adversarial networks (GANs) to augment this data, which should directly address the model’s current limitations in generalisation. Beyond faces, we could also incorporate other types of information, such as audio, text, and behaviour, to obtain a more comprehensive picture of ASD diagnosis. Factors such as speech patterns, eye movements, and body language, when combined with facial image analysis, can significantly enhance the reliability of diagnoses and lead to systems that utilise multiple types of data.

While AI shows exciting potential for diagnosing autism spectrum disorder (ASD), we must carefully address serious ethical questions. Privacy and data security are major concerns, especially with sensitive facial images and biometric information. Protecting people’s privacy is our top priority, and any future use of these methods should incorporate strong encryption techniques, such as homomorphic encryption and secure multi-party computation, to prevent data leaks. We should also consider using differential privacy to help protect individual data, and federated learning could be a suitable approach for training models without direct access to private patient information. Following data protection rules, such as GDPR and HIPAA, is crucial for maintaining patient confidentiality and ensuring that people trust AI in healthcare.

We must also be particularly cautious about bias and fairness in our AI models. AI often learns biases from the data it has been trained on, which can lead to variations in diagnostic accuracy across different groups of people. For example, if our datasets contain more male examples than female examples, the model may not perform as well in diagnosing ASD in females. This illustrates the importance of systematically checking for fairness. To address these issues, building upon the successful cross-ethnic validation demonstrated in this study, we should conduct fairness audits, employ techniques to mitigate bias, such as reweighting and adversarial debiasing, and ensure that our training data encompasses a diverse range of examples that accurately represent how ASD manifests in different genders, ethnicities, and age groups. If we want everyone to have fair access to healthcare, we must actively work to eliminate these biases in our AI models.

We also need to think carefully about transparency and making sure patients give informed consent when AI is used in their healthcare. Using AI can raise questions about whether patients have control over their own decisions and whether we can understand how the AI is making its predictions. To build trust and make these systems acceptable, it is crucial to create clear and easy-to-understand consent forms that inform people about how AI systems will use their data. We should also utilise “explainable AI” (XAI) techniques so that the AI does not just provide a “yes” or “no” answer, but also offers reasons that doctors and caregivers can understand. Furthermore, allowing doctors to adjust the sensitivity of the AI model according to a patient’s specific situation could significantly enhance the system’s utility in real-world clinics.

We also need to look beyond the technical and legal aspects and consider how AI-driven ASD screening might impact individuals and society as a whole. Automatic screening for ASD brings some risks, like people being unfairly labelled, receiving the wrong diagnosis, or relying too much on what the AI says. If the AI says that someone might have ASD when they do not, it could cause unnecessary worry, social stigma, and treatments they do not need. On the other hand, if the AI misses someone who has ASD, they might miss out on early help and support. To ensure that we avoid these problems, we need to collaborate with psychologists, doctors, and patient support groups to validate these AI tools. This will help to ensure that we use them in an ethical and medically sound manner. Additionally, AI screening should primarily be used to assist doctors in making decisions rather than making the final diagnosis itself. We need doctors to stay involved and oversee the AI’s output so we do not rely too much on the automated results.

We also need to focus on ensuring that everyone can use these AI-based ASD screening tools, thereby helping to reduce health inequalities worldwide. Often, places with fewer resources lack easy access to specialists, making it crucial to explore more affordable ways to conduct real-time screening, such as using smartphone apps. We should also consider different cultures and languages when designing these models so that they are more likely to be accepted by people from various backgrounds. Making the AI models open-source could also help more people to utilise them and allow communities to collaborate on improving them, ensuring that ASD screening solutions reach underserved populations worldwide.

Overall, our study demonstrates that utilising AI to analyse faces has significant potential for screening autism spectrum disorder (ASD), providing a promising approach to early detection that could reach a large number of people and be cost-effective. While we deliberately trained our system with a mix of people, as you would see in a real clinic, the fact that it still performed slightly better in some groups suggests that future research should try to include more female data and use special training methods that help to reduce bias, without reducing the system’s general efficacy.

## 6. Conclusions

This study represents a significant advancement in autism spectrum disorder (ASD) screening by employing deep ensemble learning and domain adaptation to analyse unseen facial images in real-time. By leveraging our clinically validated UIFID dataset alongside Kaggle and YTUIA, our proposed system effectively learns the diverse facial patterns associated with ASD, making it more adaptable and generalisable for real-world diagnosis.

Our ensemble learning approach demonstrated outstanding performance, achieving 99.6% accuracy within the same type of data, 96% across multiple different datasets, and a robust 90.6% on the UIFID dataset, highlighting its ability to handle variations in data. Furthermore, by addressing biases related to gender and age through better dataset balancing, we significantly improved the fairness of our model, particularly for females and younger individuals with autism spectrum disorder (ASD). The resulting ensemble model demonstrated a more equitable distribution of accuracy across demographic subgroups, with the accuracy for female ASD classification increasing from 78.4% to 84.2%, thereby reducing overall disparities.

Even with these positive results, we still have some hurdles to overcome. For example, there is not as much data on females with ASD in our datasets, and how ASD looks can change as people age. Additionally, we have not yet utilised some of the more advanced AI techniques for pinpointing facial features, such as vision transformers. In the future, we should focus on acquiring more diverse datasets, utilising learning techniques that adapt to different age groups, and exploring methods to combine AI models that prioritise certain features. This could make our system even more robust and accurate.

By combining AI tools with thorough clinical testing, a focus on fairness, and clear ethical guidelines, we believe that this research sets the stage for the next generation of ASD diagnostics. Our goal is to create screening solutions that are inclusive and provide equal access to early detection for everyone with autism spectrum disorder (ASD) in a world that is increasingly utilising AI.

## Figures and Tables

**Figure 1 diagnostics-15-01601-f001:**
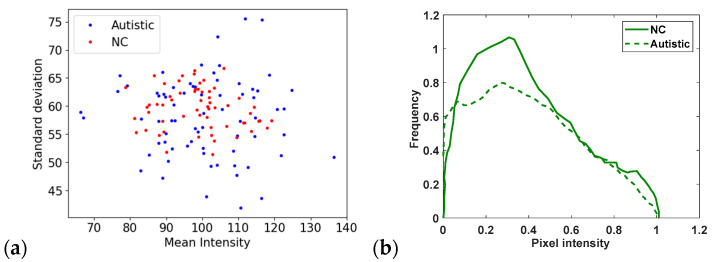
(**a**) Relationship between mean and standard deviation of pixel intensities in facial images; (**b**) distribution of normalised pixel intensities in UIFID autism classification dataset.

**Figure 2 diagnostics-15-01601-f002:**
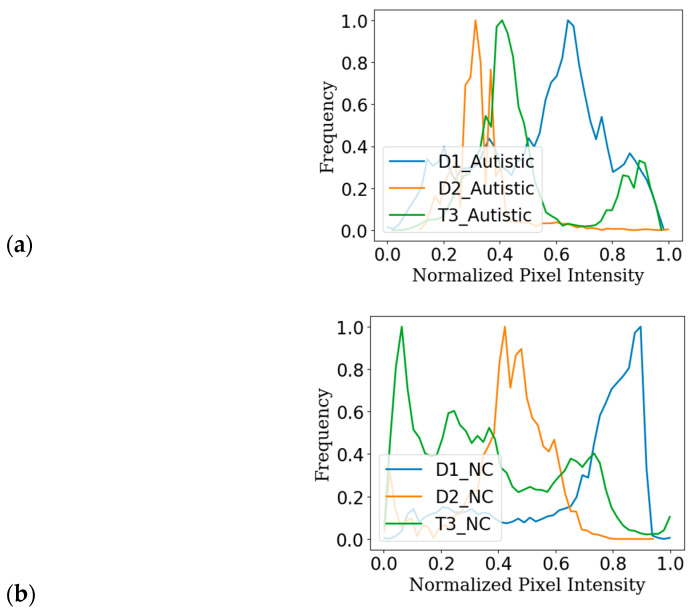
Visualisation of domain shift in image (**a**) normalized pixel intensity distributions for (**a**) only autistic samples (ASD); (**b**) only normal control (NC) samples; and (**c**) all the samples across Kaggle, YTUIA, UIFID datasets.

**Figure 3 diagnostics-15-01601-f003:**
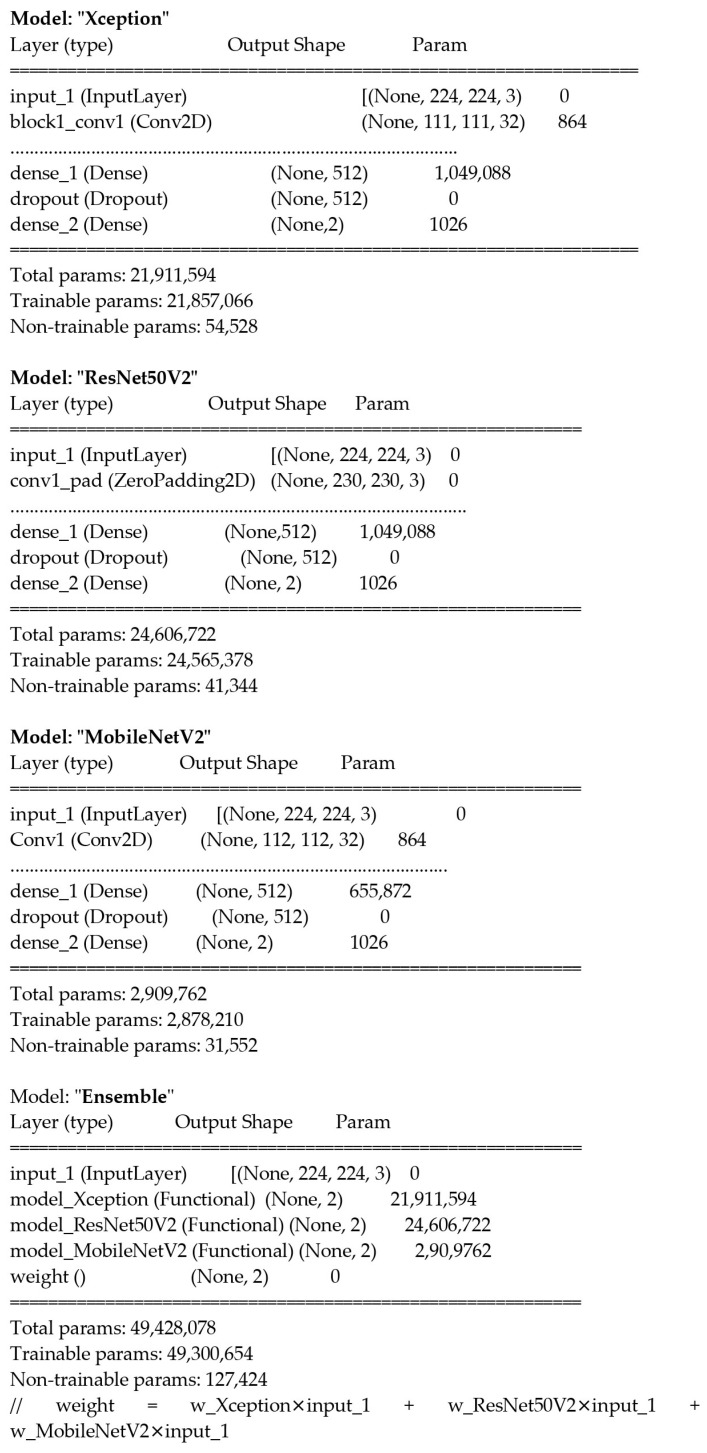
Modified CNN architectures for binary classification with tailored top layers and ensemble learning.

**Figure 4 diagnostics-15-01601-f004:**
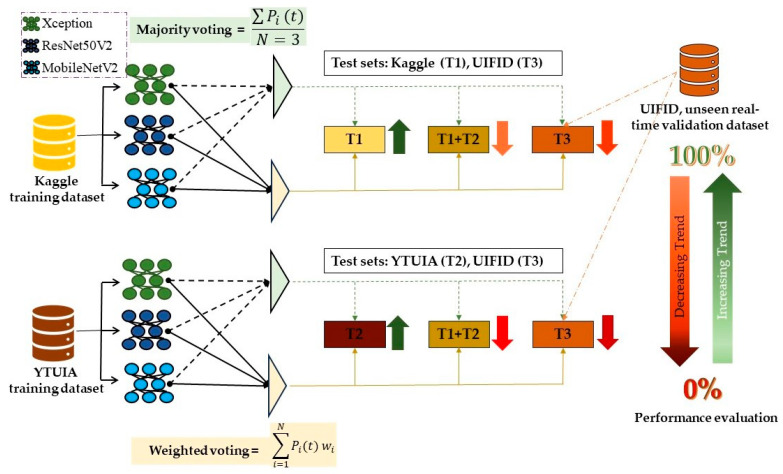
Schematic of ensemble learning with models within the same dataset.

**Figure 5 diagnostics-15-01601-f005:**
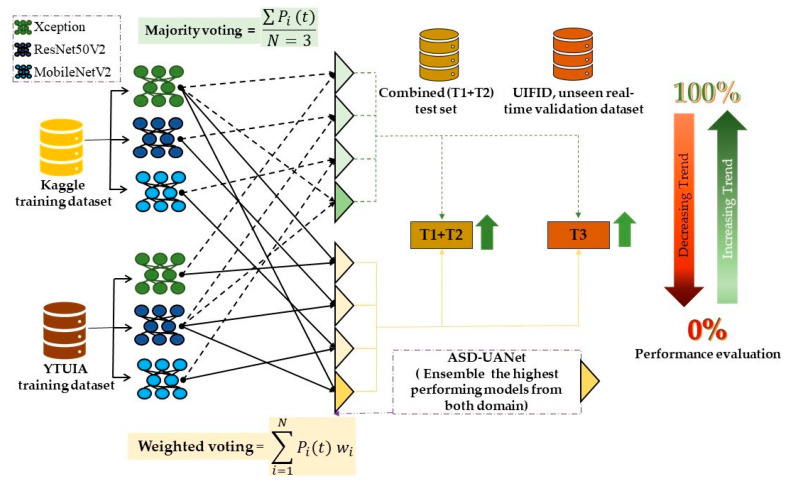
Schematic of ensemble learning with models for different datasets.

**Figure 6 diagnostics-15-01601-f006:**
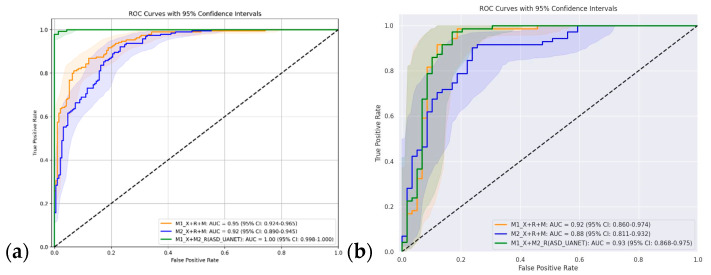
(**a**) ROC for different mixed-domain approaches (T1+T2); (**b**) ROC for different unseen-domain approaches (T3).

**Figure 7 diagnostics-15-01601-f007:**
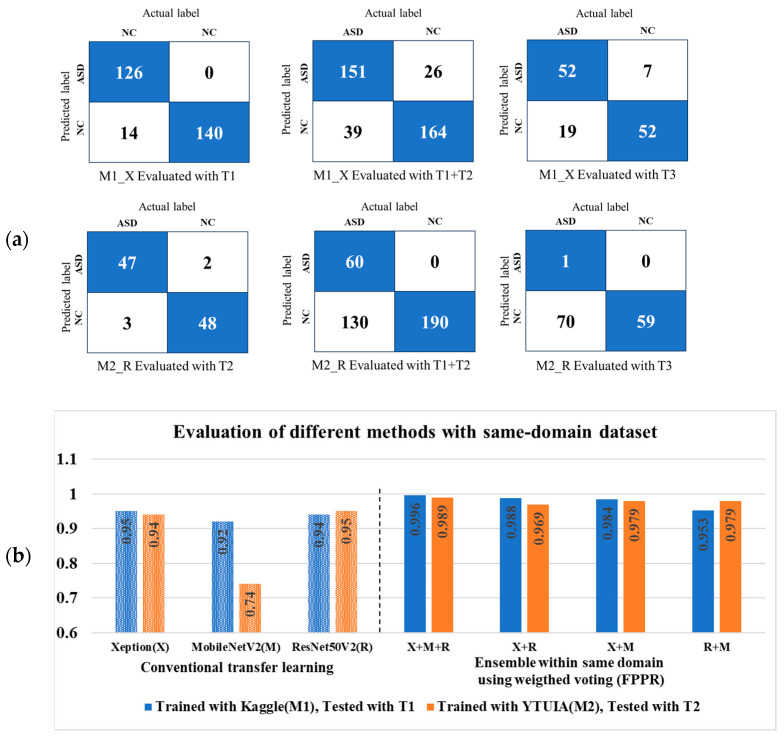
(**a**) A confusion matrix evaluating the cross-domain performance of the M1 and M2 models using standard transfer learning. The matrix highlights a pattern of frequent incorrect predictions (white cells) when models were tested on data from a different source. Correct predictions are shown in blue. (**b**) A comparison of transfer learning and ensemble learning on same-domain test datasets, evaluated using the False Positive Prediction Rate (FPPR). Across the figure, the legend indicates that lighter colors represent transfer learning, while deeper colors signify the superior performance of ensemble learning.

**Figure 8 diagnostics-15-01601-f008:**
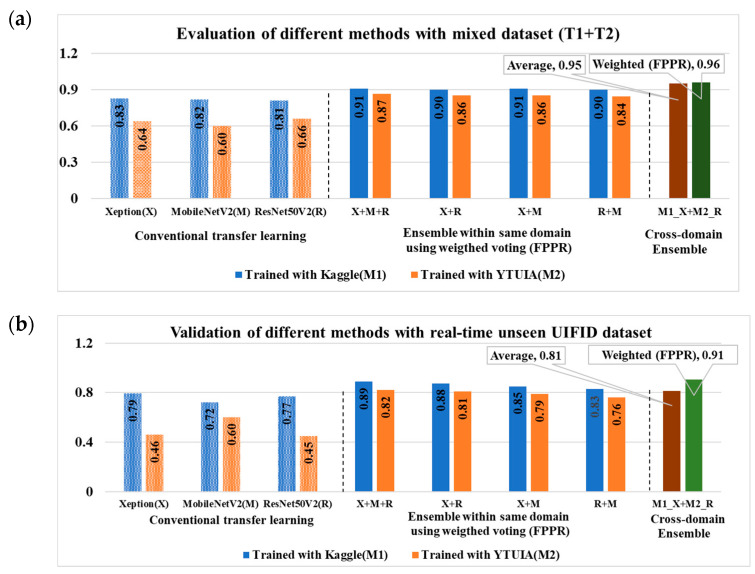
Performance evaluation of different methods for ASD screening: (**a**) evaluation of methods using the combined test set (T1+T2), demonstrating accuracy improvements of up to 0.96 using the cross-domain weighted ensemble using the FPPR; (**b**) validation of methods with the unseen real-time UIFID dataset, showing domain adaptation capabilities, where the FPPR-based cross-domain ensemble achieved the highest accuracy of 0.91. Blue bars represent the performance of models trained on the Kaggle (M1) dataset. Orange bars represent the performance of models trained on the YTUIA (M2) dataset. For the final “Cross-domain Ensemble,” the brown bar shows the result of average voting, and the green bar shows the result of the higher-performing, FPPR-weighted voting.

**Figure 9 diagnostics-15-01601-f009:**
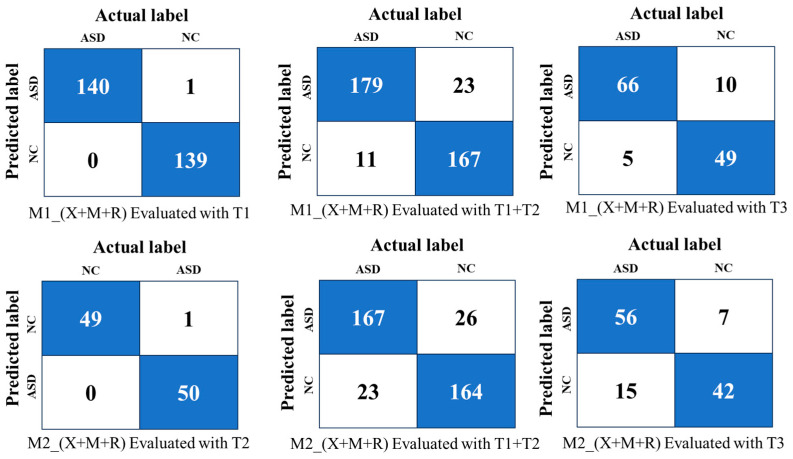
Confusion matrices illustrating the performance of the FPPR-based ensemble of all three models, which achieved the best results in ASD screening across different datasets and domains. The matrices provide detailed predictions for true positives, true negatives, false positives, and false negatives. Dark Blue (Main Diagonal): Represents the number of correct predictions (True Positives and True Negatives). White (Off-Diagonal): Represents the number of incorrect predictions (False Positives and False Negatives).

**Figure 10 diagnostics-15-01601-f010:**
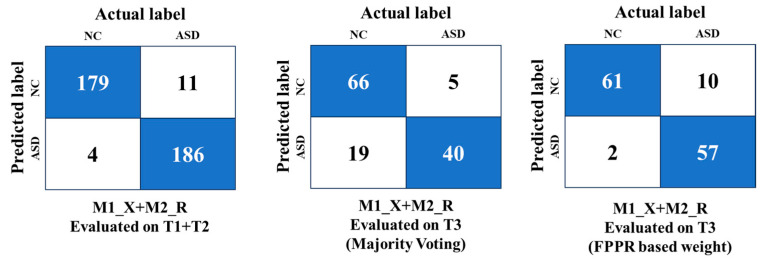
Confusion matrices showing the performance of the proposed cross-domain ensemble model (ASD-UANet) for ASD screening. In each matrix, the dark blue cells on the main diagonal highlight the number of correct predictions (true positives and true negatives), while the white off-diagonal cells indicate the incorrect predictions (false positives and false negatives).

**Table 1 diagnostics-15-01601-t001:** Recent research on ASD screening using ensemble learning.

Ref.	Multiple Datasets	Algorithms	Domain Adaptation	Reported Accuracy
[21]	No (speech data)	SVM, KNN	No	93.00
[22]	No (ABIDE)	Xception, VGG16	No	87.00
[23]	Yes (Kaggle ASD, FER2013, own eye gaze data)	Own CNN	No	84.00
[24]	No (eye gaze)	ResNet	No	66.00
[25]	No (kaggle ASD)	MobileNet, EfficientNet, Xception	No	80.00
[26]	Yes (kaggle ASD, FADC)	ViT, SVM	No	99.81

**Table 2 diagnostics-15-01601-t002:** Summary of demographic and clinical information.

Dataset	Total Samples	Male–Female Ratio	Age Range (Years)	ASD–NC Ratio	Notes
Kaggle	3014	3:1	2–8	1:1	Freely available online [29]
YTUIA	1168	3:1	1–11	1:1	Freely available online [30]
UIFID	130	2:1	3–11	3:2	Used only for validation [31]

**Table 3 diagnostics-15-01601-t003:** (**a**). Details of the UIFID Facial Image Dataset. (**b**). UIFID Facial Image Dataset (before and after balancing).

(**a**)								
**Class**	**Age (Years)**					**Sample Population**		
	**Minimum**	**Maximum**	**Standard dev**		**Average**	**Female**	**Male**	**Total**
Autistic (ASD)	3	12	2.4		8.03	7	24	31
Normal (NC)	5	8	1		6.95	10	10	20
(**b**)								
**Class**	**Male**		**Female**	**Age (2–6) Male**	**Age (7–11) Male**	**Age (2–6) Female**		**Age (7–11)** **Female**
Before Balancing (130 samples)								
ASD	24		7	5	19	6		1
NC	10		10	2	8	4		6
After Balancing (118 samples)								
ASD	19		7	5	14	6		1
NC	10		10	2	8	4		6

**Table 4 diagnostics-15-01601-t004:** Dataset splits and distribution across training, testing, and validation sets for binary classification.

Split	Kaggle	YTUIA	UIFID	Binary Class
Training set	2654 (D1)	1068 (D2)		0—non-ASD 1—ASD
Testing set	280 (T1)	100 (T2)	130 (T3)	
Validation set	80	-		

**Table 5 diagnostics-15-01601-t005:** Combination of same-domain and cross-domain models for ensemble learning.

Sl No	Same-Domain Models	Cross-Domain Models
1	Xception + MobileNetV2 + ResNet50V2	M1-Xception + M2-Xception
2	Xception + MobileNetV2	M1-ResNet50V2 + M2-ResNet50V2
3	Xception + ResNet50V2	M1-MobileNetV2 + M2-MobileNetV2
4	ResNet50V2 + MobileNetV2	M1-Xception + M2-ResNet50V2

**Table 6 diagnostics-15-01601-t006:** Performance comparison of models trained on different datasets across evaluation sets (transfer learning).

Algorithm	Model	Evaluated on	Accuracy	AUC	Precision	Recall	F1-Score
M1 (Trained with D1)	Xception	T1	0.95	0.98	0.95	0.95	0.95
		T1+T2	0.83	0.9	0.83	0.83	0.83
		T3	0.792	0.882	0.792	0.792	0.792
	MobileNetV2	T1	0.92	0.96	0.92	0.92	0.92
		T1+T2	0.82	0.86	0.82	0.82	0.82
		T3	0.723	0.801	0.723	0.723	0.723
	ResNet50V2	T1	0.94	0.96	0.94	0.94	0.94
		T1+T2	0.81	0.87	0.81	0.81	0.81
		T3	0.769	0.84	0.769	0.769	0.769
M2 (Trained with D2)	Xception	T2	0.94	0.98	0.94	0.94	0.94
		T1+T2	0.64	0.66	0.64	0.64	0.64
		T3	0.46	0.53	0.46	0.46	0.46
	MobileNetV2	T2	0.74	0.83	0.74	0.74	0.74
		T1+T2	0.6	0.64	0.6	0.6	0.6
		T3	0.6	0.67	0.6	0.6	0.6
	ResNet50V2	T2	0.95	0.96	0.95	0.95	0.95
		T1+T2	0.66	0.69	0.66	0.66	0.66
		T3	0.45	0.5	0.45	0.45	0.45

**Table 7 diagnostics-15-01601-t007:** Performance evaluation of model ensemble trained using Kaggle dataset.

Ensemble Configurations M1 (Trained with D1)	Majority Voting					Weighted Ensemble				
	Accuracy	AUC	Precision	Recall	F1-Score	Accuracy	AUC	Precision	Recall	F1-Score
	Evaluated on T1									
Xception + MobileNetV2 + ResNet50V2	0.957	0.994	0.957	0.957	0.957	0.996	0.999	0.996	0.996	0.996
Xception + MobileNetV2	0.988	0.999	0.988	0.988	0.988	0.988	0.999	0.988	0.988	0.988
Xception + ResNet50V2	0.969	0.991	0.969	0.969	0.969	0.984	0.994	0.984	0.984	0.984
ResNet50V2 + MobileNetV2	0.984	0.997	0.984	0.984	0.984	0.953	0.996	0.953	0.953	0.953
	Evaluated on T1+T2									
Xception + MobileNetV2 + ResNet50V2	0.867	0.943	0.867	0.867	0.867	0.91	0.945	0.91	0.91	0.91
Xception + MobileNetV2	0.879	0.925	0.879	0.879	0.879	0.901	0.944	0.901	0.901	0.901
Xception + ResNet50V2	0.897	0.94	0.897	0.897	0.897	0.909	0.956	0.909	0.909	0.909
ResNet50V2 + MobileNetV2	0.845	0.947	0.845	0.845	0.845	0.901	0.945	0.901	0.901	0.901
	Evaluated on T3									
Xception + MobileNetV2 + ResNet50V2	0.796	0.86	0.796	0.796	0.796	0.82	0.89	0.82	0.82	0.82
Xception + MobileNetV2	0.766	0.86	0.766	0.766	0.766	0.81	0.875	0.81	0.81	0.81
Xception + ResNet50V2	0.8	0.88	0.8	0.8	0.8	0.79	0.851	0.79	0.79	0.79
ResNet50V2 + MobileNetV2	0.758	0.78	0.758	0.758	0.758	0.76	0.828	0.76	0.76	0.76

**Table 8 diagnostics-15-01601-t008:** Performance evaluation of model ensemble trained using YTUIA dataset.

Ensemble Configurations M2 (Trained with D2)	Majority Voting					Weighted Ensemble				
	Accuracy	AUC	Precision	Recall	F1-Score	Accuracy	AUC	Precision	Recall	F1-Score
	Evaluated on T1									
Xception + MobileNetV2 + ResNet50V2	0.958	0.979	0.958	0.958	0.958	0.989	0.986	0.989	0.989	0.989
Xception + MobileNetV2	0.958	0.982	0.958	0.958	0.958	0.969	0.987	0.969	0.969	0.969
Xception + ResNet50V2	0.969	0.996	0.969	0.969	0.969	0.979	0.988	0.979	0.979	0.979
ResNet50V2 + MobileNetV2	0.906	0.968	0.906	0.906	0.906	0.979	0.988	0.979	0.979	0.979
	Evaluated on T1+T2									
Xception + MobileNetV2 + ResNet50V2	0.787	0.871	0.787	0.787	0.787	0.869	0.921	0.869	0.869	0.869
Xception + MobileNetV2	0.81	0.901	0.81	0.81	0.81	0.855	0.916	0.855	0.855	0.855
Xception + ResNet50V2	0.855	0.9	0.855	0.855	0.855	0.855	0.915	0.855	0.855	0.855
ResNet50V2 + MobileNetV2	0.81	0.899	0.81	0.81	0.81	0.844	0.918	0.844	0.844	0.844
	Evaluated on T3									
Xception + MobileNetV2 + ResNet50V2	0.797	0.864	0.797	0.797	0.797	0.82	0.877	0.82	0.82	0.82
Xception + MobileNetV2	0.758	0.818	0.758	0.758	0.758	0.81	0.839	0.81	0.81	0.81
Xception + ResNet50V2	0.719	0.789	0.719	0.719	0.719	0.79	0.817	0.79	0.79	0.79
ResNet50V2 + MobileNetV2	0.687	0.74	0.687	0.687	0.687	0.76	0.802	0.76	0.76	0.76

**Table 9 diagnostics-15-01601-t009:** Performance evaluation of cross-domain model ensemble (trained with both Kaggle and YTUIA).

Ensemble of Models (M1 Trained on D1 and M2 Trained on D2)	Majority Voting					Weighted Ensemble				
	Accuracy	AUC	Precision	Recall	F1-Score	Accuracy	AUC	Precision	Recall	F1-Score
	Evaluated on T1+T2									
(M1+M2) Xception	0.957	0.990	0.957	0.957	0.957	0.950	0.991	0.950	0.950	0.950
(M1+M2) ResNet50V2	0.910	0.976	0.910	0.910	0.910	0.943	0.989	0.943	0.943	0.943
(M1+M2) MobileNetV2	0.912	0.961	0.912	0.912	0.912	0.946	0.978	0.946	0.946	0.946
M1_Xception + M2_ResNet50V2 (ASD-UANet)	0.951	0.990	0.951	0.951	0.951	0.960	0.990	0.960	0.960	0.960
	Evaluated on T3									
Xception + MobileNetV2 + ResNet50V2	0.813	0.881	0.813	0.813	0.813	0.867	0.921	0.867	0.867	0.867
Xception + MobileNetV2	0.742	0.796	0.742	0.742	0.742	0.750	0.801	0.750	0.750	0.750
Xception + ResNet50V2	0.742	0.831	0.742	0.742	0.742	0.860	0.920	0.860	0.860	0.860
ResNet50V2 + MobileNetV2	0.812	0.877	0.812	0.812	0.812	0.906	0.930	0.906	0.906	0.906

**Table 10 diagnostics-15-01601-t010:** Recent methods of domain adaption for ASD diagnosis.

Ref.	Method	Algorithm	Training/Validation Dataset	Accuracy
[27]	Active learning (using w12)	MobileNetV2	Kaggle + YTUIA/UIFID	0.828
[46]	Dataset federation	Xception	Kaggle + YTUIA/UIFID	0.680
[47]	Data-centric approach (Align only)	Xception	Kaggle/UIFID	0.823
Current work	Proposed weighted Ensemble	ASD-UANet	Kaggle + YTUIA/UIFID	0.906

**Table 11 diagnostics-15-01601-t011:** Comparison of recent studies on ensemble learning methods for ASD detection, highlighting datasets, algorithms, accuracies, and improvements over baseline approaches.

Ref.	Algorithms	Accuracy	Improvement (%)	Numbers and Names of Datasets	Validation Dataset
[22]	Xception, VGG16, own CNN	87.00	0.5	01 (ABIDE)	Same-domain (15%)
[23]	Own CNN	84.00	7.5	03 (Kaggle ASD, FER2013, eye gaze data)	Same-domain (17%)
[24]	ResNet	66.00	6.3	01 (eye gaze)	Same-domain
[25]	MobileNet, EfficientNet, and Xception	80.00	4.9	01 (kaggle ASD)	Same-domain (10%)
[26]	ASD-EVNet	99.81	2.7	02 (kaggle ASD, FADC)	Same-domain (20%)
A proposed method for domain adaptation with ensemble learning					
Current work	Xception + MobileNetV2 + ResNet50V2	99.60	4.6	02 (kaggle ASD, YTUIA)	Same-domain (T1)
Current work	(ASD-UANet)	96.0	30	02 (kaggle ASD, YTUIA)	Cross-domain (T1+T2)
Current work	(ASD-UANet)	90.6	30	02 (kaggle ASD, YTUIA)	UIFID (T3)

**Table 12 diagnostics-15-01601-t012:** Model performance across different subgroups.

Model	Male (G1)	Male (G2)	Female (G1)	Female (G2)	Overall Accuracy
M1_X	86.50%	91.20%	79.80%	83.50%	88.30%
M2_R	84.10%	89.50%	81.20%	82.70%	86.90%
M1_X+M2_R	87.30%	90.80%	80.50%	84.20%	89.10%

## Data Availability

The data presented in this study are published here: https://www.mdpi.com/2075-4418/14/6/629 (accessed on 19 June 2025).

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
