# Peer review of "Robust Autism Spectrum Disorder Screening Based on Facial Images (For Disability Diagnosis): A Domain-Adaptive Deep Ensemble Approach"

_diagnostics, 2025, doi:10.3390/diagnostics15131601_

Round 1
Reviewer 1 Report
Comments and Suggestions for Authors
Dear authors,
Thank you for the opportunity to review your study. Your work tackles an important gap—establishing a clinically useful, image-based screening pipeline for ASD that can generalize across acquisition domains. I was particularly impressed by (i) your effort to curate the UIFID validation set and (ii) the attempt to formalize a weighting heuristic (“Fifty Percent Priority Rule”) for the ensemble. Below I summarise the principal points that, in my view, must be addressed before the manuscript can proceed. Where feasible I have grouped remarks under headings.
1 Introduction & Literature Context
- Duplication – Several passages (e.g., lines 188-194, 286-307) repeat earlier sentences nearly verbatim. Please condense to avoid redundancy and maintain reader focus.
2 Methods
- Dataset ethics & consent
- Specify how consent was obtained for publicly scraped YouTube (YTUIA) frames and clarify licence terms for Kaggle images.
- UIFID approval is referenced, but the consent process for minors (parents/guardians) should be described explicitly.
- Data-split logic
- Explain the “subject-independent” split in precise numbers (how many subjects, how many images per fold).
- Provide a diagram or table summarising training/validation/test partitions across all datasets.
- Hyper-parameters
- Learning-rate schedules, weight-decay, optimiser settings, and early-stopping criteria are not reported in Section 3.5. These are required for reproducibility.
- Statistical testing
- You cite Welch–Satterthwaite t-tests and χ² analyses but do not provide exact P-values for each comparison. Include test statistics, degrees of freedom, and correction for multiple comparisons (if any).
3 Results
- Baseline comparisons
- Add at least one modern vision-transformer baseline (e.g., Swin-V2, ViT-B/16 fine-tuned) to contextualise the ensemble gains.
- ROC/AUC plots
- Supply ROC curves with 95 % CIs for each evaluation set; current tables list single-point AUCs only.
- Bias analysis
- The gender-wise accuracy gap (≈ 7 %) deserves deeper analysis. Consider adding stratified confusion matrices and discussion of potential mitigation strategies.
4 Discussion & Interpretation
- Over-claiming – The conclusion states that your system “achieves state-of-the-art performance.” Please temper this claim unless you benchmark against SOTA methods on the same splits.
- Clinical workflow – Outline a plausible screening scenario (camera type, inference time, hand-off to clinician) to illustrate real-world feasibility.
Numerous run-on sentences, tense shifts, and informal phrases (“you know”, “huge difference”) weaken the academic tone. A professional copy-edit is recommended. Pay special attention to subject-verb agreement (“data are”, not “data is”) and consistency in terminology (e.g., “ASD-NC ratio” vs “ASD:NC”).
Reviewer 2 Report
Comments and Suggestions for Authors
Dear authors,
your paper is really interesting, dealing with the use of CNNs in ASD diagnosis. AI use in diagnostic field is a really important scientific area to deepen.
Some revisions are suggested in order to improve the overall quality of the paper.
- It would be better to open the introduction with the definition of ASD as reported by the DSM-5. Lines 40-45 attempt to report this information, but in a less than scientific way.
- Lines 47-50: when you report lack of clear biological markers to diagnose ASD, it is true, but it should be rewritten in a more scientific way. For example: "Despite scientific evidence found out that..." (see 10.1038/s41576-020-0231-2).
- Referring to clinical checklist, there are some assessment tools really valid and reliable (see ADOS).
- Lines 73-74: Is there any evidence in literature about the benefits of this technique? If yes, report that. If no, please comment this lacking.
- Lines 79: What researchers? Please, add references.
- A brief paragraph to introduce what is CNN and its role would be essential to the reader.
- How did you do literature review? Has it been a systematic search? Please describe the methods used.
- A lot of information into the introduction are unnecessary. It would be better to combine the introduction and literature review into a more fluid paragraph.
- Write better the study aims.
- Lines 188-195 are unnecessary.
- Lines 547-553 are unnecessary.
- Into the discussion there are very few references to the existing literature.
- Paper is too long, hardly to read and the organization of the part are not clear.
Reviewer 3 Report
Comments and Suggestions for Authors
Dear Authors,
The work done and the presentation are good. Anyhow, the following comments need to be addressed to improve the quality of the manuscript.
- Abstract needs revision. Include the quantitative details of the results.
- Including more recent works from the year 2025 in the literature is recommended.
- Include the details of the training set and test set proportions.
- Table 6 needs to be moved to the next page. It is recommended to include the F1-score details in this table. Because Precision and Recall individually do not support the performance evaluation.
- Discuss the details of the difficulties when implementing cross-domain.
Reviewer 4 Report
Comments and Suggestions for Authors
1. What is the reason for specifically choosing pre-trained MobileNetV2, ResNet50V2, and Xception models? Additionally, an ensemble model composed of these three models should not be considered an innovation. The novelty seems very weak. Also, please rewrite the contributions of this study by taking into account the contributions of these models.
2. Including models such as DenseNet121-201, ConvNeXt, VGGNet, and EfficientNet in the tables would be beneficial.
3. F1 score values need to be added to Table 6.
4. Precision, recall, and F1 score values should be added to Tables 7, 8, and 9.
Round 2
Reviewer 1 Report
Comments and Suggestions for Authors
The authors have responded thoughtfully to most of the substantive critiques. Ethical documentation, and the addition of confidence intervals represent significant improvements. Remaining concerns are largely clarifications rather than fundamental flaws.
Recommendation: I would now be comfortable moving to minor revision focused on providing the small language fixes, and a more explicit discussion of cross-ethnic performance statistics.
Author Response
Response: We sincerely thank the reviewer for their positive feedback and for the suggestion to provide a more explicit discussion of cross-ethnic performance. We agree this is a crucial point, and our study was designed specifically to ensure the model's robustness across diverse populations. We have now clarified this methodology and its results in the manuscript.
Our strategy is as follows:
- Baseline Training: We begin by training on the Kaggle dataset. As noted by previous research, this dataset is predominantly composed of White children (89%), which presents a potential for ethnic bias.
- Bias Mitigation: To counteract this, we incorporate the YTUIA dataset, which includes a broader and more balanced spectrum of ethnicities. This step is critical for exposing the model to a more diverse representation of facial features during training, enhancing its ability to learn more generalizable patterns.
- Cross-Ethnic Validation: The model's generalizability is then rigorously tested on the UIFID dataset, which was collected exclusively from a single, distinct ethnic group (Bangladeshi) not prominently featured in the initial training data. Our proposed ASD-UANet ensemble model achieved high accuracy (90.6%) on this unseen dataset, demonstrating its effectiveness in a different ethnic context.
- Multi-Ethnic Evaluation: Finally, the model's robustness is confirmed on a mixed test set combining samples from both Kaggle and YTUIA. The high accuracy achieved here further underscores the model’s successful generalization across varied ethnicities.
By training on a combination of biased and diverse data and validating on a distinct ethnic group, our framework demonstrates strong cross-ethnic performance. We have now added this explicit discussion to Section 5.6 of the manuscript.
Reviewer 2 Report
Comments and Suggestions for Authors
Dear Authors,
Thank you for your valuable work in revising the manuscript and for carefully considering all the suggestions I previously provided.
Some minor revisions are still needed:
-
Lines 131–146 are not necessary. In scientific papers, it is generally not required to summarize how the paper is organized.
-
I understand the rationale behind including a literature review section. However, not following a standardized methodology may introduce significant bias in the way data are reported. Moreover, although you described the search strategy in the point-by-point response letter, this procedure is not mentioned in the manuscript itself.
I remain unconvinced about your approach on this point. Nonetheless, since you consider it appropriate and relevant, I kindly ask that you apply a more rigorous and methodologically sound process. I leave the final decision to the editor.
Author Response
Response: Thank you for your continued valuable feedback and for carefully reviewing our revised manuscript. We appreciate the opportunity to make further improvements based on your suggestions.
We have addressed the two minor revisions you have pointed out:
- Summary of Paper Organization: We agree with your assessment that the paragraph summarizing the paper's organization is not essential for a scientific article. In accordance with your suggestion, we have removed this section (the text corresponding to lines 131–146 in your review) to improve the conciseness of the introduction.
- Literature Review Methodology: We thank you for this critical point. We agree completely that for methodological transparency and to reduce potential bias, the search strategy for the literature review must be described within the manuscript itself. We apologize for this oversight in our previous revision.
To rectify this, we have now incorporated a new introductory paragraph at the beginning of Section 2 (Literature Review). This new text explicitly details the search strategy, including the databases searched, the specific keywords used for the search (e.g., "Autism Spectrum Disorder," "ensemble learning," "deep learning," "facial image," "domain adaptation"), and the criteria used for selecting the studies cited in our review. We believe this addition now makes our process rigorous, methodologically sound, and transparent to the reader, as you rightly suggested.
We are grateful for your guidance, which has significantly strengthened our manuscript. We hope that these revisions now fully address your concerns.
Reviewer 4 Report
Comments and Suggestions for Authors
I thank the authors for doing what was requested.
Author Response
Response: We thank the reviewer for their positive confirmation and for the time and effort they invested in reviewing our manuscript.